# Structural and functional insights of the human peroxisomal ABC transporter ALDP

**Yutian Jia[1†], Yanming Zhang[1†], Wenhao Wang[1], Jianlin Lei[2], Zhengxin Ying[1]\*, Guanghui Yang[1]\***

[1]State Key Laboratory for Agrobiotechnology, Department of Nutrition and Health, College of Biological Sciences, China Agricultural University, Beijing, China; [2]Technology Center for Protein Sciences, Ministry of Education Key Laboratory of Protein Sciences, School of Life Sciences, Tsinghua University, Beijing, China

**Abstract** Adrenoleukodystrophy protein (ALDP) is responsible for the transport of very-long-chain fatty acids (VLCFAs) and corresponding CoA-esters across the peroxisomal membrane. Dysfunction of ALDP leads to peroxisomal metabolic disorder exemplified by X-linked adrenoleukodystrophy (ALD). Hundreds of ALD-causing mutations have been identified on ALDP. However, the pathogenic mechanisms of these mutations are restricted to clinical description due to limited structural and biochemical characterization. Here we report the cryo-electron microscopy structure of human ALDP with nominal resolution at 3.4 Å. ALDP exhibits a cytosolic-facing conformation. Compared to other lipid ATP-binding cassette transporters, ALDP has two substrate binding cavities formed by the transmembrane domains. Such structural organization may be suitable for the coordination of VLCFAs. Based on the structure, we performed integrative analysis of the cellular trafficking, protein thermostability, ATP hydrolysis, and the transport activity of representative mutations. These results provide a framework for understanding the working mechanism of ALDP and pathogenic roles of disease-associated mutations.

**\*For correspondence:**
yingzhengxin@cau.edu.cn (ZY);
guanghuiyang@cau.edu.cn (GY)

[†]These authors contributed equally to this work

**Competing interest:** The authors declare that no competing interests exist.

## Editor's evaluation

The adrenoleukodystrophy protein (ALDP) or ABCD1 is an ABC transporter that participates in the transport of free very long-chain fatty acids and their CoA esters across the peroxisomal membrane. By determining the cryo EM structure of human ABCD1 the study represents a valuable insight into its transport mechanism and the mechanistic basis for mutations causing the severe neurodegenerative disorder, X-linked adrenoleukodystrophy. The structure and functional studies of disease-causing mutations are solid and will appeal to the transporter and medical genetics communities.

## Introduction

X-linked adrenoleukodystrophy (ALD) is an inherited disease characterized by progressive demyelination of the central nervous system (*Berger et al., 2014*). Loss of myelin slows down the transmission of nerve impulses and triggers neuroinflammation (*Ferrer et al., 2010*; *Kettwig et al., 2021*; *Singh et al., 2009*). Pathogenesis of ALD is tightly associated with mutations on the adrenoleukodystrophy protein (ALDP), which transports very-long-chain fatty acid-CoAs (VLCFA-CoAs) from cytosol into the peroxisome (*Figure 1*; *Mosser et al., 1993*). Over 900 disease-derived mutations have been identified on ALDP (https://adrenoleukodystrophy.info/mutations-biochemistry/mutations-biochemistry). ALDP, also known as ABCD1, belongs to the ATP-binding cassette sub-family D. The other three members

**Figure 1.** Functional and structural characterization of human adrenoleukodystrophy protein (ALDP). (**A**) Schematic of very-long-chain fatty acids transport into the peroxisomes by ALDP. (**B–C**) ATP hydrolysis of ALDP stimulated by C22:0-CoA (data are represented as mean ± SD; n=3; three biological repeats). (**D**) Cryo-electron microscopy structure of ALDP. The transmembrane domains (TMDs) and nucleotide-binding domains (NBDs) are indicated.

The online version of this article includes the following source data and figure supplement(s) for figure 1:

**Source data 1.** Cryo-electron microscopy data collection, refinement, and validation statistics.

**Figure supplement 1.** Cryo-electron microscopy (cryo-EM) analysis of adrenoleukodystrophy protein (ALDP).

**Figure supplement 2.** Cryo-electron microscopy (cryo-EM) density of human adrenoleukodystrophy protein (ALDP).

**Figure supplement 3.** Comparison of unknown densities in the ligand-free structure with the different substrates.

**Figure supplement 4.** Structural comparison of all members of ATP-binding cassette sub-family D (ABCD) transporters in human.

are ALDP-related protein (ALDRP/ABCD2; *Lombard-Platet et al., 1996*), PMP70/ABCD3 (*Kamijo et al., 1990*), and a cobalamin transporter ABCD4 (*Coelho et al., 2012*). ABCD1–3 are distributed on peroxisomes, while ABCD4 is localized on lysosomes (*Xu et al., 2019*). ALDP and ALDRP transport VLCFA with different specificities (*van Roermund et al., 2011*). ALDRP may compensate for the function loss of ALDP (*Holzinger et al., 1997*; *Liu et al., 1999*; *Lombard-Platet et al., 1996*). PMP70 was reported to transport LCFA-CoA (*Ranea-Robles et al., 2021*; *van Roermund et al., 2014*).

Structure-based functional characterization of ALDP is eagerly required for understanding the VLCFA transport mechanism and the pathogenic roles of ALD-derived mutations. To address these questions, we first performed ATP hydrolysis assay using purified ALDP. In the presence of C22:0-CoA, ALDP exhibits robust ATP hydrolysis with the $V_{max}$ of ~193 ± 8.2 mol Pi min$^{-1}$ mol$^{-1}$ protein and $K_m$ value of ~0.17 µM C22:0-CoA (*Figure 1B and C*). The structure of human ALDP reveals an assembly of two identical subunits that exhibit a domain-swapped arrangement (*Figure 1D* and *Figure 1—figure supplements 1–2* and *Figure 1—source data 1*). Each subunit contains six transmembrane

helices (TMs; *Figure 1D* and *Figure 1—figure supplement 2*). A short helix located at the peroxi-somal side (extracellular helix [EH]) is a featured structural element of ALDP based on sequence alignment (*Figure 1—figure supplement 4*). Two coiled-coil domain-like densities at the C-terminus were observed (C-terminal helix [CH]). The local resolution of these densities is estimated to be ~4.0 Å, which is not accurate to build an atomic model.

During submission of this study, four other groups independently reported ALDP structures in different states (*Chen et al., 2022*; *Le et al., 2022*; *Wang et al., 2022*; *Li et al., 2021*). Our structure is the ligand-free state as no substrates were supplemented during purification. We observed two unknown densities at the crevice of TMDs, which were also reported in the ligand-free structure by *Chen et al., 2022*. When superimposed to the oleoyl-CoA- or C22:0-CoA-bound structures, the unknown densities partially overlap with the substrates (*Figure 1—figure supplement 3*). Such observation indicates a probable extrusion of the unknown densities during substrate loading. Structure comparison of our structure with the C22:0-CoA-bound and ATP-bound structure reveals dramatic conformational changes during the transport cycle. For instance, the distance between TM4 and TM6 at the juxta-membrane region changes from 23.6 Å to 13.1 Å upon substrate binding and further gets

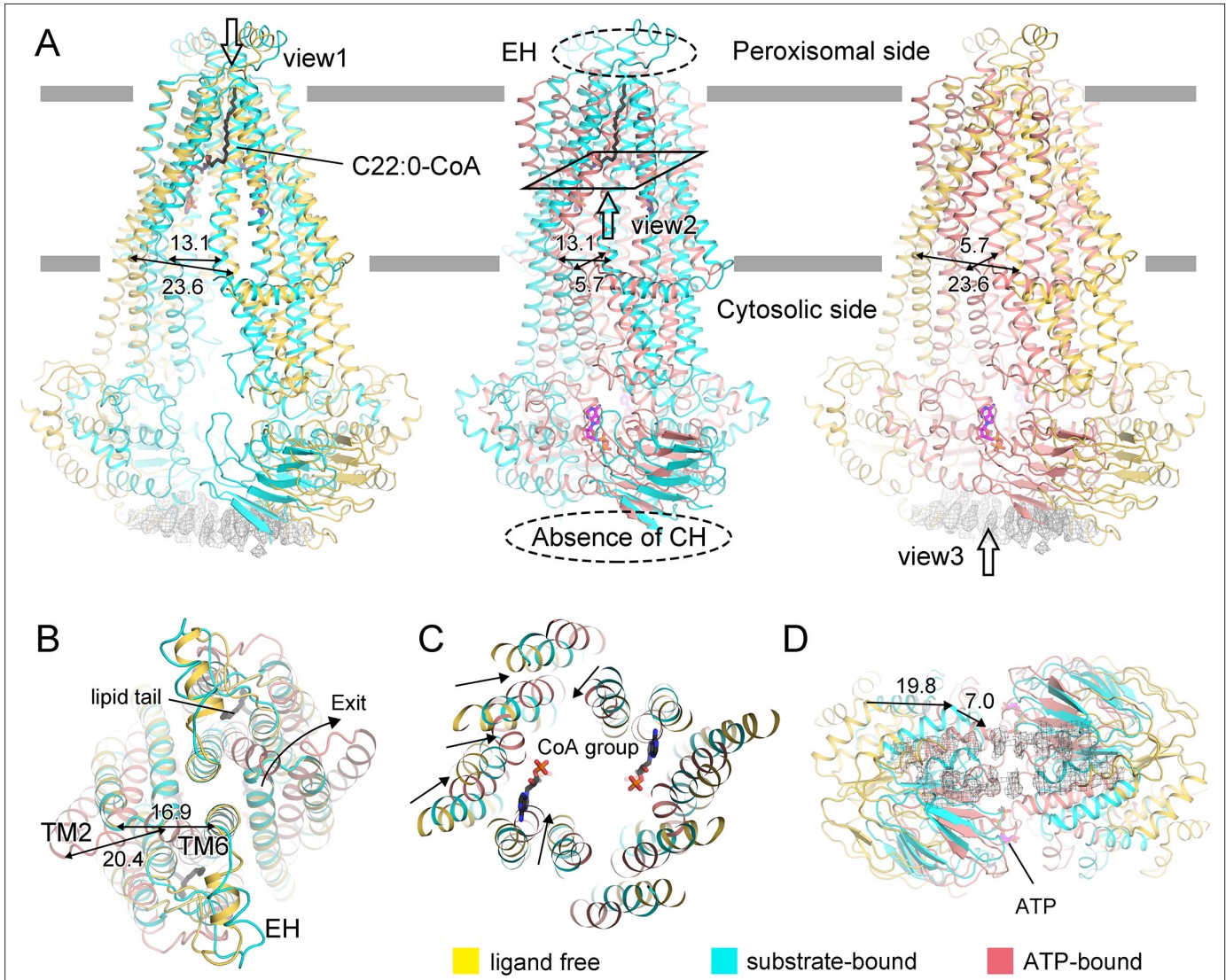

**Figure 2.** Conformational changes of adrenoleukodystrophy protein (ALDP) during the whole transport cycle. (**A**) Structural alignments of ALDP in ligand-free, C22:0-CoA-bound (7VZB) (*Chen et al., 2022*), and ATP-bound states (7SHM) (*Wang et al., 2022*). The distance change (in Å) between transmembrane helix 4 (TM4) and TM6 reflects the conformational changes during lipid transport. The hollow arrows indicate three views of ALDP that are enlarged in **B–D**. CH: C-terminal helix; EH: extracellular helix.

closed to 5.7 Å in the presence of ATP. The nucleotide-binding domains (NBDs) undergo continuously spatial rearrangement. Consequently, the TM4-6 and TM1-2 bend to open the peroxisomal side exit for substrate (*Figure 2*). Conformational change of TMDs is accompanied with the movement of NBDs and the disappearance of EH or CH.

Structure determination of ALDP allows classification of ALD-associated mutations. 970 ALD-associated mutations in ALDP affecting 232 residues have been reported. Among these 232 residues, 88 harbor mutations to two or more types of amino acids, which we defined as hotspot residues. Based on the structural mapping, the 88 hotspot residues can be roughly classified into 3 groups (*Figure 3B*). The first group of 10 residues lines along the substrate-binding cavity. These residues likely play a role in substrate coordination and delivery. The second group of 35 residues is located in other region of the TMDs, which undergoes conformational changes. The third group of 43 residues is located on the NBDs that may have influence on ATP hydrolysis (*Figure 3B* and *Figure 3—figure supplement 1*).

To examine how ALD-associated mutations affect the function of ALDP, we selected those reported to have normal expressed levels but dysfunction for integrative analysis (*Coll et al., 2005*; *Feigenbaum et al., 1996*; *Guimarães et al., 2002*). These mutations include W339R, S342P, G343V, A396T, Q544R, and T693M (*Imamura et al., 1997*; *Kemp et al., 2001*; *Lan et al., 2011*; *Liu et al., 2022*; *Pan et al., 2005*; *Takahashi et al., 2007*; *Watkins et al., 1995*; *Wichers et al., 1999*). In addition, to evaluate the role of different structure elements, we designed four constructs based on structural changes. The S164/Y310 residues on TMDs, anchored by a hydrogen bond, both move quite large distance during the transport cycle but remain relatively static distance between themselves. The E291/R518 residues are mapped to the interface between TMD and NBD. The side chain of E291 stretches into the NBD of another protomer, while R518 binds to the α-phosphate of ATP. These two pairs of residues represent two kinds of dimeric interfaces. We mutated these residues into alanine to examine how disturbance of the dimeric assembly affects the activity. Another two constructs are truncation of EH (Δ364–374) or CH (Δ683–745).

We first measured the VLCFA-CoA transport. Different mutations are individually expressed in HEK293 cells. The cytosolic C22:0-CoA, C24:0-CoA, and C26:0-CoA were measured by quantitative mass spectrometry (*Wang et al., 2019*), and the transport ability was calculated (*Figure 3C*). Because the substrate transport is coupled with ATP hydrolysis, we also measured the ATPase activity of these mutants using purified proteins (*Figure 3C*). To exclude the possibility that the mutants may alter the cellular localization and thermostability compared to wild-type (WT), we investigated their subcellular localization and measured the $T_m$ values of purified proteins. Compared with A95D, a disease-causing mutation known to not colocalize with peroxisome and reduce the C24:0 β-oxidation (*Morita et al., 2013*), all the selected mutants show puncta distribution and colocalize with catalase, suggesting their normal trafficking to the peroxisomes (*Figure 3—figure supplement 2*). The similar $T_m$ values of purified mutant proteins rule out the instability-caused functional abrogation (*Figure 3—figure supplement 3*).

The transport activity and the ATP hydrolysis of the mutants are normalized to WT. All six disease-derived mutants exhibit decreased transport toward three species of VLCFA-CoA. Five mutants show abrogated ATP hydrolysis except for the mutation S342P. The ALD mutation T693M is mapped on the CH. Because the CH cannot be observed in the ATP-bound state, to evaluate the potential pathogenic role of the T693M mutant, we generated the homomeric model of ALDP by AlphaFold (hereafter as AF-model) (*Evans et al., 2022*; *Jumper et al., 2021*). The predicted model of ALDP displays similar conformation to the ATP-bound state. Albeit the part of CH in AF-model needs further validation, the prediction may provide potential clues for interpreting the T693M mutation at the current stage. In the AF-model, T693 is located near W664, which is surrounded by ALD mutants on the NBD. The distance between the side chains of T693 and W664 can reach in 3.4 Å (*Figure 3—figure supplement 4*). Replacement of Thr by Met residue with large side chain may affect the neighboring residues near W664 to disturb the ATP hydrolysis. For the four designed constructs, alanine substitution of S164/Y310 and E291/R518 has negligible influence on the ATP hydrolysis. By contrast, the lipid transport decreased because destabilization of the interaction between these residues may disturb the conformational changes during substrate translocation. As one featured structural element, deletion of the EH (Δ364–374) may generate structural constraints between TM5 and TM6 during the conformational change of ALDP (*Chen et al., 2022*; *Le et al., 2022*; *Wang et al., 2022*). Truncation of CH (Δ683–745)

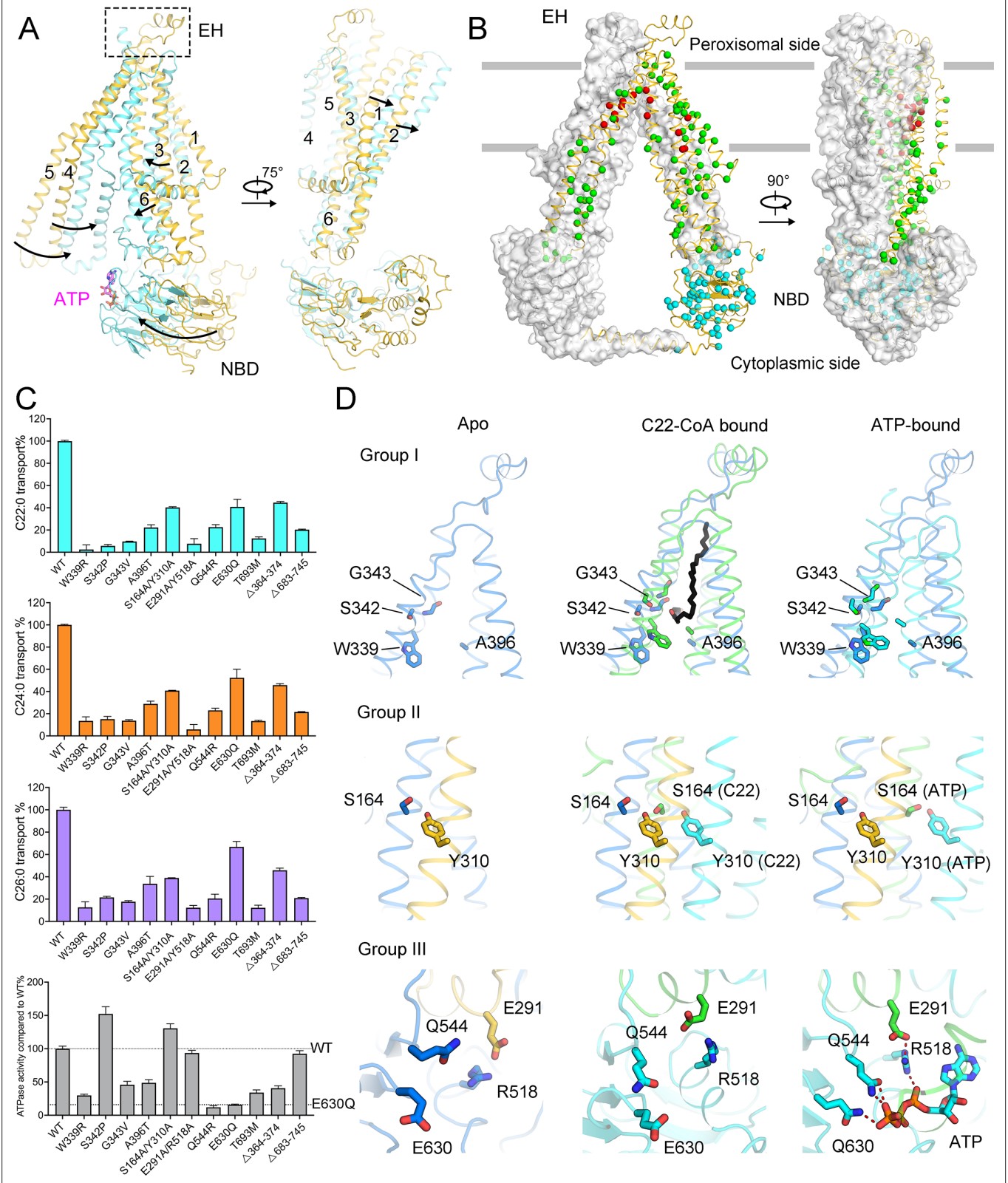

**Figure 3.** Functional analysis of adrenoleukodystrophy (ALD)-associated mutations. (**A**) Conformational changes of adrenoleukodystrophy protein (ALDP) from ligand-free to ATP-bound state. (**B**) Structural mapping of clinical-derived pathogenic mutations of ALDP. The Cα atoms of the indicated residues are shown as spheres. The residues involved in substrate binding are shown as red spheres. Other mutation sites on transmembrane domains are presented in green. Mutations on nucleotide-binding domain (NBD) and the single-mutation T693M on C-terminal helix are colored cyan,

*Figure 3 continued on next page*

*Figure 3 continued*

respectively. (**C**) Spatial location of selected ALD mutants and rationally designed mutations. (**D**) Transport of cytosolic C22:0-CoA, C24:0-CoA, C26:0-CoA, and ATP hydrolysis by wild-type (WT) ALDP and mutations (data are represented as mean ± SD; n = 3; three biological repeats). EH: extracellular helix.

The online version of this article includes the following source data and figure supplement(s) for figure 3:

**Figure supplement 1.** Structural mapping of residues that harbor recurring adrenoleukodystrophy-associated mutants.

**Figure supplement 2.** Cellular localization of representative mutations.

**Figure supplement 3.** Expression level and thermo-stability of representative mutations.

**Figure supplement 3—source data 1.** Western blot for *Figure 3—figure supplement 3A*.

**Figure supplement 4.** Structural comparison between cryo-electron microscopy (EM) structure and the predicted one by Alphafold (AF).

**Figure supplement 5.** Comparison between adrenoleukodystrophy protein (ALDP) and phosphatidylcholine transporter ABCB4.

does not change the assembly of NBD but affects the conformational changes between the two protomers, thus the ATP hydrolysis level slightly decreased with abrogated lipid transport.

To gain insights into the transport of different lipids by ABC transporters, we compared ALDP with the phospholipid transporter ABCB4 (*Nosol et al., 2021*; *Olsen et al., 2020*). Typically, ALDP harbors two substrate-binding pockets that expand perpendicularly to the membrane, but ABCB4 binds one phosphatidylcholine (PC) (*Figure 3—figure supplement 5*). The dimension of one binding pocket is ~1695 Å$^3$ for C22:0-CoA and ~1236 Å$^3$ for PC calculated by ProteinPlus (*Schöning-Stierand et al., 2020*). The different formulas between VLCFA-CoA and PC may provide clues for the different cavities. Regardless of the flexibility, the length of C22:0-CoA (~55 Å) is larger than PC (~40 Å) (*Figure 3—figure supplement 5B*). The PC molecule has two acyl tails that can be coordinated by ABCB4. The VLCFAs have only one hydrophobic tail. More residues in a larger binding pocket of ALDP may help in stabilizing the flexible long tail of VLCFA-CoA. The EH and CH facilitate conformational changes to expand the binding pocket. Taken together, the integrative characterization of ALDP provides a framework to illustrate the molecular basis of VLCFA transport and the function of ALD-derived mutations.

## Methods
### Protein expression and purification

The coding sequence of human *ALDP* was subcloned into the pFastBac1 vector with a N-terminal Flag tag. The recombinant *ALDP* was expressed using the Bac-to-bac system (Invitrogen). Briefly, bacmids were generated in DH10Bac competent cells. The resulting baculoviruses were amplified in Sf9 insect cells (Invitrogen). ALDP was overexpressed in Sf9 insect cells grown in the serum-free medium (Gibco). 60 hr after P3 virus infection, transfected cell pellet was collected and homogenized in the buffer containing 25 mM Tris-HCl, pH 7.4, 150 mM NaCl supplemented with 1 mM phenylmethylsulfonyl fluoride, 1.3 µg ml$^{-1}$ aprotinin, 0.7 µg ml$^{-1}$ pepstatin, and 5 µg ml$^{-1}$ leupeptin. After brief sonication, the suspension was supplemented with 2,2-didecylpropane-1,3-bis-β-D-maltopyranoside (LMNG, Anatrace) to a final concentration of 1% (w/v) and cholesteryl hemisuccinate Tris salt (CHS, Anatrace) to 0.1% (w/v). After incubation at 4°C for 2 hr, the mixture was centrifuged at 150,000 g for 30 min. The supernatant was mixed with anti-Flag M2 affinity gel (Sigma). The gel was rinsed with 25 mM Tris-HCl, pH 7.4, 150 mM NaCl, 0.005% LMNG, and 0.0005% CHS. The target protein was eluted with wash buffer plus 200 µg ml$^{-1}$ flag peptide and further purified through gel-filtration chromatography in the buffer containing 25 mM Tris-HCl, pH 7.4, 150 mM NaCl, and 0.06% digitonin. Peak fractions were concentrated for cryo-EM analysis.

### Preparation of the cryo-EM samples

Aliquots of 4 µl ALDP was dropped onto glow-discharged holey carbon grids (Quantifoil Au R1.2/1.3, 300 mesh). The grids were blotted for 3 s and flash-frozen in liquid ethane using Vitrobot Mark IV (FEI). The sample was imaged on an FEI 300 kV Titan Krios transmission electron microscope equipped with a Cs corrector and Gatan GIF Quantum energy filter (slit width 20 eV), recorded by a Gatan K2 Summit detector with a nominal magnification of ×640,000. A series of defocus values from –1.5 to –1.8 µm was used during data collection. Each image was dose-fractionated to 32 frames with a total electron

dose of ~50 e⁻ Å⁻² and a total exposure time of 5.6 s. AutoEMation II (developed by J. Lei) (*Lei and Frank, 2005*) was used for fully automated data collection. All stacks were motion-corrected using MotionCor2 with a binning factor of 2 (*Zheng et al., 2017*), resulting in a pixel size of 1.0979 Å. The defocus values were estimated using Gctf (*Zhang, 2016*), and dose weighing was performed concurrently (*Grant and Grigorieff, 2015*).

## Cryo-EM data processing and model building

In total, 1,259,972 particles were auto-picked from these 1922 movie stacks using Gautomatch (developed by Kai Zhang, http://www.mrc-lmb.cam.ac.uk/kzhang/Gautomatch) (*Figure 1—figure supplement 1*). After two-dimensional classification, 472,168 particles were selected to generate the initial model with a mask diameter of 200 Å in C1 symmetry. To avoid model bias, the initial model was low-pass filtered for the following three-dimensional (3D) classification. The 472,168 particles were subjected to 50 iterations of global angular search 3D classification. Each of the 50 iterations has one class and a step size of 7.5°. For each of the last five iterations (iterations 46–50) of the global search, the local angular search 3D classification was executed with a class number of 3, a step size of 3.75°, and a local search range of 15°. A total of 289,321 particles were selected from the local angular searching 3D classification after removing the redundant particles and particles from bad classes (*Figure 1—figure supplement 1*). The selected particles were subjected to 3D autorefinement, yielding a density map with average resolution at 3.6 Å. After multi-reference 3D classification, 135,662 particles were selected for autorefinement with C2 symmetry, resulting a map at 3.4 Å resolution. The structure of ABCD4 was used as a template for the model building of ALDP. The structure was refined in real space using PHENIX with secondary structure and geometry restraints (*Adams et al., 2002*). The atomic model was manually improved using COOT (*Emsley and Cowtan, 2004*).

## Determination of cytosolic very-long-chain fatty acids

Sample preparation and liquid chromatography-mass spectrometry (LC-MS/MS) conditions as described (*Wang et al., 2019*). Chromatographic separation was achieved using Q Exactive HF LC-MS/MS system and Phenomenex Luna 5 µm C5 column (i.d. 100×2.0 mm). Mobile phase A contained HPLC-grade $H_2O$-ACN 40/60 (v/v) with 10 mM ammonium acetate, and mobile B was isopropanol-ACN 90/10 (v/v). Both WT *ALDP* and mutant constructs were subcloned into a pCAG vector and expressed in HEK293F cells (Invitrogen). Transfected cells were harvested after 60 hr. $2.0×10^7$ cells were collected and washed by PBS. 1 mL ACN and 10 µL internal standard mix solution were added to samples or quality control sample (expressed by empty vector). After sonication and centrifugation, 50 µL hydrochloric acid (12 mol/L) was added into the supernatant then VLCFA was hydrolyzed at 70°C for 1 hr. Each sample was extracted with 2 ml hexane, the dried residue was reconstituted in 100 µL methanol. 5 µL aliquots of the reconstitutes were loaded onto the LC–MS/MS system for analysis.

## Immunofluorescence cytochemistry

Hela cells fixed in 4% Paraformaldehyde (20 min) were permeabilized with 0.1% Triton X-100 in PBS (30 min, room temperature), blocked with SuperBlock (30 min), before incubation with rabbit anti-catalase antibody (1:500; Abcam, ab16731). Immunoreactivity was visualized by AlexaFluor-488-conjugated goat anti-rabbit IgG (1:1000; 1 hr at room temperature).

## ATPase activity assay

For determining ATP hydrolysis capacity, wtABCD1 and mutants were deleted N-terminal 54 residues to improve stability. The malachite green ATPase assay was chosen to exhibit the ABCD1 ATPase activity in the presence of C22:0-CoA (behenoyl coenzyme A, ammonium salt) as previously described (*Baykov et al., 1988*). Purified ABCD1 in LMNG/CHS micelles was pre-incubated on ice for 10 min in 20 mM Tris-HCl, pH 7.5, 50 mM KCl, 1 mM DTT, 2 mM $MgCl_2$ with different concentrations of C22:0-CoA. The 100 µl reaction started by supplementing with 2 mM ATP and then carried out at 37°C for 30 min, after that terminated by adding 25 µl fresh Gold-mix solution (1 mM malachite green, 1.2% ammonium molybdate, and 0.15% Tween 20). The mixture was incubated at RT for 30 min before being detected absorbance in 96-well micro-plate at 630 nm. Statistical analysis was performed using GraphPad Prism 7.

## The thermal stability assay

The thermal stability assay was performed by measuring intrinsic tryptophan fluorescence and light back-scattering for fragment screening by a Prometheus NT.48 device (NanoTemper Technologies GmbH, Munich, Germany) (*Ahmad et al., 2021*). wtABCD1 and mutations were purified in LMNG/CHS and concentrated to 0.5 mg/ml, after centrifuged at 14,000 × g for 15 min at 4°C, supernatant was loaded with standard capillaries (NanoTemper Technologies GmbH, Munich, Germany; Cat# PR-C002) into the Prometheus device and subjected to a linear thermal ramp (1°C/min, from 25 to 95°C) and collected fluorescence at 350 nm and 330 nm. Unfolding transition midpoints were determined automatically from the first derivative of the fluorescence ratio (F350/F330). Data analysis was performed using PR.Therm Control (NanoTemper Technologies GmbH) and GraphPad Prism 7.

## Acknowledgements

We thank the Tsinghua University Branch of China National Center for Protein Sciences (Beijing) for the cryo-EM facility and the computational facility support, Dr. Xiaomin Li, Dr. Fan Yang and Tao Liu for technical support in EM data acquisition. We thank Mr. Wei Gao and Ms. Yang Yue at Mo-In Biotechnology (Beijing) Co., Ltd for the data acquisition and analysis of the protein thermostability. This work was supported by National Natural Science Foundation of China (3217110084; 32130081; 32100759); Chinese Universities Scientific Fund (15050004; 15050017; 15051002); and Young Elite Scientists Sponsorship Program by China Association for Science and Technology.

## Additional information

### Funding

| Funder | Grant reference number | Author |
|---|---|---|
| National Natural Science Foundation of China | 3217110084 | Guanghui Yang |
| National Natural Science Foundation of China | 32130081 | Guanghui Yang |
| National Natural Science Foundation of China | 32100759 | Zhengxin Ying |
| Chinese Universities Scientific Fund | 15050004 | Guanghui Yang |
| Chinese Universities Scientific Fund | 15051002 | Guanghui Yang |
| Chinese Universities Scientific Fund | 15050017 | Guanghui Yang |

The funders had no role in study design, data collection and interpretation, or the decision to submit the work for publication.

### Author contributions

Yutian Jia, Yanming Zhang, Wenhao Wang, Data curation, Formal analysis, Investigation; Jianlin Lei, Software; Zhengxin Ying, Resources, Formal analysis, Investigation; Guanghui Yang, Conceptualization, Resources, Data curation, Software, Formal analysis, Supervision, Funding acquisition, Validation, Investigation, Visualization, Methodology, Writing - original draft, Project administration, Writing - review and editing

### Author ORCIDs

Yutian Jia ⓘ http://orcid.org/0000-0001-6921-7222
Jianlin Lei ⓘ http://orcid.org/0000-0002-9384-8742
Zhengxin Ying ⓘ http://orcid.org/0000-0003-0948-4948
Guanghui Yang ⓘ http://orcid.org/0000-0002-6835-1611

Decision letter and Author response
Decision letter https://doi.org/10.7554/eLife.75039.sa1
Author response https://doi.org/10.7554/eLife.75039.sa2

## Additional files

### Supplementary files
• Transparent reporting form

### Data availability
Cryo-EM data have been deposited in PDB under the accession code 7VR1. All data generated or analysed during this study are included in the manuscript and supporting files. Source data files have been provided for Figure 1 and Figure 3—figure supplement 3.

The following dataset was generated:

| Author(s) | Year | Dataset title | Dataset URL | Database and Identifier |
|---|---|---|---|---|
| Yang GH, Jia YT, Zhang YM | 2022 | Cryo-EM structure of the ATP-binding cassette sub-family D member 1 from *Homo sapiens* | https://www.rcsb.org/structure/7VR1 | RCSB Protein Data Bank, 7VR1 |

The following previously published dataset was used:

| Author(s) | Year | Dataset title | Dataset URL | Database and Identifier |
|---|---|---|---|---|
| Kemp S | 2022 | The ABCD1 Variant Database | https://adrenoleukodystrophy.info/mutations-and-variants-in-abcd1 | The ABCD1 Variant Database, abcd1 |

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
