## [Editor Report]

The adrenoleukodystrophy protein (ALDP) or ABCD1 is an ABC transporter that participates in the transport of free very long-chain fatty acids and their CoA esters across the peroxisomal membrane. By determining the cryo EM structure of human ABCD1 the study represents a valuable insight into its transport mechanism and the mechanistic basis for mutations causing the severe neurodegenerative disorder, X-linked adrenoleukodystrophy. The structure and functional studies of disease-causing mutations are solid and will appeal to the transporter and medical genetics communities.

---

## [Decision Letter]

**Decision letter after peer review:**

Thank you for submitting your article "Structure insights of the human peroxisomal ABC transporter ALDP" for consideration by *eLife*. Your article has been reviewed by 3 peer reviewers, including David Drew as Reviewing Editor and Reviewer #1, and the evaluation has been overseen by Kenton Swartz as the Senior Editor. The following individual involved in review of your submission have agreed to reveal their identity: Konstantinos Beis (Reviewer #3).

Essential revisions:

1. The modelled two putative Co-A esters lack the positively charged residues for coordinating phosphates. Indeed, the Co-A binding in the human ABCD1 structure by Chen and co-workers make more chemical sense https://www.biorxiv.org/content/10.1101/2021.09.24.461565v2.full.pdf. This study found they same shaped density as you have seen in your ABCD1 structure, but have modelled the lipid 18:0 Lyso PE molecule into this location instead. Judging the deposited maps, we think that this is the more likely lipid for your density, rather than CoA as proposed. We need to see an update of the substrate-binding site and the corresponding mutations to either the new substrate-binding site position as in the competing paper or more solid experimental data for keeping with the currently modelled substrate-binding site.

2. Certain assumptions have been made but there is no references or experimental validation from this work to warrant such conclusions; eg p6 'Pathogenic mutations on these residues may disturb the stability of ALDP. The residues of the third group are located on the NBDs, which may hinder the hydrolysis of nucleotides. Given the explosion of structural information on ABC transporters (for example, of the ~50 human ABC transporters, structures have been reported for the human A1, A4, B1, B2, B3, B4, B6, B8, B10, B11, C7, D4, G2, G5, G8 transporters in addition to D1), it is a challenge to interpret new results in the context of existing knowledge about ABC transporters. The transport assays provide an approach to connecting structure and function, but too little information is provided about the assays to make those connections. Of the ~900 mutants, why were the variants in figure 4B selected? Where are they located in the structure? What do the results of the transport assay mean? For example, is the reduction of C22:S-CoA in the cytosol for the G343V mutant due to impaired transport or to reduced expression, misfolding or some other cause? mutations are mainly mapped to the TMDs, which may affect the conformational changes during substrate transport.

Please show the correct trafficking, expression (normalised to WT) and folding of all mutations tested in HEK293 cells.

Please show that the purified protein is capable of turning over ATP and that the basal ATPase activity stimulated by the substrate. We would recommend carrying out complementary ATPase assays on key mutations also. It is critical to better integrate the mutagenesis data with the structure.

3. Please provide some rationale basis for the additional structural features found in ABCD1 (peroxisomal helix and the C-terminal coil-coiled domain) and how this might be related to its function?

4. The final model is difficult to follow and this, of course, relates to the "actual" location of the substate-binding site. Is an outward-facing conformation to be expected in ABCD1? Furthermore, the conformational changes as proposed by comparing to AlphaFold models should be made with caution until experimental validation of these states. Please incorporate the AlphaFold models more carefully into the paper, i.e.,what are the state-specific co-evolved residue-pairs predicted in the AlphaFold models, do these make sense, and can these contact pairs be experimentally validated by mutagenesis?

5. The paper needs to be extensively re-written. We would like to see how the structure uncovers new mechanistic insights. With the additional functional data requested, consider how the ABCD1 structure enables a understanding of why very long chain fatty acids use such an ABC transporter structures for being imported into peroxisomes. How does this differ to other ABC transporters that transport lipids and why?

*Reviewer #1 (Recommendations for the authors):*

1. The modelled two putative Co-A esters lack the positively charged residues for coordinating phosphates. Indeed, the Co-A binding in the human ABCD1 structure by Chen and co-workers make more chemical sense https://www.biorxiv.org/content/10.1101/2021.09.24.461565v2.full.pdf. This study found they same shaped density as you seen in your ABCD1 structure, but have modelled the lipid 18:0 Lyso PE molecule into this location instead. Judging the deposited maps I think that this is the more likely lipid for your density, rather than CoA as proposed. Please read this submission more carefully and update accordingly as I find it worrying that this has not been picked up.

2. Please check that ABCD1 mutations are trafficked correctly to peroxisomes. Also one needs to normalise uptake against expression level. Please complement substrate uptake experiments with ATPase activity measurements also.

3. This paper is difficult to follow in places and using AlphaFold structures to make conclusions regarding local conformational changes seems premature at this stage without further validation.

4. Please provide some rationale basis for the additional structural features found in ABCD1 (peroxisomal helix and the C-terminal coil-coiled domain) and how this might be related to its function?

*Reviewer #3 (Recommendations for the authors):*

The CoA-ester binding should be discussed in more details.

Although several mutants have been prepared, they are not really discussed in the context of the manuscript or previously published work. The discussion needs to expand further. Certain assumptions have been made but there is no references or experimental validation from this work to warrant such conclusions; eg p6 'Pathogenic mutations on these residues may disturb the stability of ALDP. The residues of the third group are located on the NBDs, which may hinder the hydrolysis of nucleotides. The other mutations are mainly mapped to the TMDs, which may affect the conformational changes during substrate transport.'. Based on their structure it should be possible to explain better why and how the mutants may affect the function.

The authors should show that the purified protein is capable of turning over ATP and that the basal ATPase activity stimulated by the substrate.

Figure 4B; are all mutants expressing at similar levels as the WT protein? They authors should show a Western Blot. Additionally, what do the error bars represent?

The authors should briefly state in the M and M how they prepared the Alphafold models; ie collar or full Alphafold. Why not use the Alphafold multimer and make the homodimer?

[Editors’ note: further revisions were suggested prior to acceptance, as described below.]

Thank you for resubmitting your work entitled "Structural and functional insights of the human peroxisomal ABC transporter ALDP" for further consideration by *eLife*. Your revised article has been evaluated by Kenton Swartz (Senior Editor) and a Reviewing Editor.

The manuscript has been improved but there are some remaining issues that need to be addressed, as outlined below:

– The main impact of the paper is the ALDP structure with a number of supporting mutations. The current Manuscript word length is around 1800 words and 4 figures, which is close to the requirements for a short report (1500 words and 3 figures). We think the paper would come across much better as a short report, and ask that you please re-write to meet these requirements and improve the readability of the text in the process.

https://reviewer.elifesciences.org/author-guide/types

The referees have further concerns to be addressed:

– line 100: "The TM2 serves as a mechanical lever: the cytosolic side moves towards the central cavity, while the peroxisomal side may move backwards the symmetry axis"

Two points:

i. is TM2 really a mechanical level? A lever is a rigid object with a fulcrum point that amplifies an input force to become an output force. Is this what is meant? If so, where is the fulcrum and what are the input and output forces?

ii. I don't understand what is being described in the second part of this sentence.

– Line 148 "Intriguingly, there are four constructs exhibit unaffected or little enhanced ATPase hydrolysis but obstructed lipid transport"

– Line 164 " Truncation of the CH does not change the completeness of the NBD"

What is meant by the completeness of the NBD?

– Line 194 – what precisely do these dimensions (42 Å or 40 Å) describe? – this implies that the TMDs have similar dimensions to accommodate their substrates – but lines 203 (and the abstract) state that the binding pocket of ALDP is much larger than ABCB4. Please clarify.

What are the volumes of the binding cavities in ALDP and ABCB4, and how does the volume of PC compare to the volume of the VLCFA substrate?

– Line 264 – Figure 4 – which PDBs are which? (ie ligand-free, substrate-bound, ATP-bound)?

– One last point that likely reflects an issue on my end, but I was not able to see the EM map postrun9.mrc in coot 0.9.6 – Coot opened the map and provided a histogram of the density, but it didn't display anything over the "A_fit_real_space_refined-coot-34_real_space_refined_001_all_states.pdb" coordinates.

– Why is the structure in line 190 compared to ABCB4 and not the other structures listed in line 90? It makes more sense to discuss the mechanism based on the available structures of the same family in addition to the ABCB4.

– It is still unclear how they selected their mutants; lines 122-125 do not have any refs of previously identified mutations. Do we know any of them being specific disease related?

– Lines 85-91 contradict each other; they state that there is lipid-like density but at the same time they write that they have an apo structure.

*Reviewer #2 (Recommendations for the authors):*

The structure of human ALDP is of significance given the role of this transporter in the transport of VLCFAs across the peroxisomal membrane. Nearly a thousand ALD-causing mutations have been identified for ALDP and the structure provides a framework for assessing their functional consequences. In this revised manuscript, the authors have addressed many of the reviewers' concerns in the rebuttal letter. Given the importance of this transporter, particularly the human version of ALDP, I am sympathetic to the publication.

The manuscript, however, needs extensive polishing to be suitable for publication, however. Here are a few examples

line 100: "The TM2 serves as a mechanical lever: the cytosolic side moves towards the central cavity, while the peroxisomal side may move backwards the symmetry axis"

Two points:

i. is TM2 really a mechanical level? A lever is a rigid object with a fulcrum point that amplifies an input force to become an output force. Is this what is meant? If so, where is the fulcrum and what are the input and output forces?

ii. I don't understand what is being described in the second part of this sentence.

Line 148 "Intriguingly, there are four constructs exhibit unaffected or little enhanced ATPase hydrolysis but obstructed lipid transport"

Line 164 " Truncation of the CH does not change the completeness of the NBD"

What is meant by the completeness of the NBD?

line 194 – what precisely do these dimensions (42 Å or 40 Å) describe? – this implies that the TMDs have similar dimensions to accommodate their substrates – but lines 203 (and the abstract) state that the binding pocket of ALDP is much larger than ABCB4. Please clarify.

What are the volumes of the binding cavities in ALDP and ABCB4, and how does the volume of PC compare to the volume of the VLCFA substrate?

line 264 – Figure 4 – which PDBs are which? (ie ligand-free, substrate-bound, ATP-bound)?

One last point that likely reflects an issue on my end, but I was not able to see the EM map postrun9.mrc in coot 0.9.6 – Coot opened the map and provided a histogram of the density, but it didn't display anything over the "A_fit_real_space_refined-coot-34_real_space_refined_001_all_states.pdb" coordinates.

*Reviewer #3 (Recommendations for the authors):*

The authors have significantly improved the manuscript with the addition of additional functional data of several mutants.

A few more points to strengthen the manuscript:

Why is the structure in line 190 compared to ABCB4 and not the other structures listed in line 90? It makes more sense to discuss the mechanism based on the available structures of the same family in addition to the ABCB4.

It is still unclear how they selected their mutants; lines 122-125 do not have any refs of previously identified mutations. Do we know any of them being specific disease related?

Lines 85-91 contradict each other; they state that there is lipid-like density but at the same time they write that they have an apo structure.

The grammar in several sections needs to be improved.

---

## [Author Response]

Essential revisions:1. The modelled two putative Co-A esters lack the positively charged residues for coordinating phosphates. Indeed, the Co-A binding in the human ABCD1 structure by Chen and co-workers make more chemical sense https://www.biorxiv.org/content/10.1101/2021.09.24.461565v2.full.pdf. This study found they same shaped density as you have seen in your ABCD1 structure, but have modelled the lipid 18:0 Lyso PE molecule into this location instead. Judging the deposited maps, we think that this is the more likely lipid for your density, rather than CoA as proposed. We need to see an update of the substrate-binding site and the corresponding mutations to either the new substrate-binding site position as in the competing paper or more solid experimental data for keeping with the currently modelled substrate-binding site.

Point accepted. We have removed the description of the potential binding site as we originally proposed to avoid controversial interpretation on the binding site of VLCFA-CoA.

We compared all the reported structures of ABCD1. Wang et al. described the oleoyl-CoA bound structure. Chen and co-workers present the C22-CoA bound structure. Xiong et al. reported the C26 and C26-CoA bound conformation. Judged from the released structure of the oleoyl-CoA or C22:0-CoA bound ABCD1 and the figure of C26:0-CoA bound ABCD1 from Xiong et al. The lipid density in our model partially overlaps with the CoA group of oleoyl-CoA and C26:0-CoA and the long tail of C22:0-CoA. Among the many tail-like densities surrounding the transmembrane region of ALDP. The density we observed that inserted into the crevice of TMDs is relatively strong. This indicates an endogenous lipid may partially occupy the large binding cavity for the VLCFA-CoA and may provide explanations for the impaired lipid transport by the mutations around the cleft mentioned in our original manuscript.

**Author response image 1. sa2fig1:** 

2. Certain assumptions have been made but there is no references or experimental validation from this work to warrant such conclusions; eg p6 'Pathogenic mutations on these residues may disturb the stability of ALDP. The residues of the third group are located on the NBDs, which may hinder the hydrolysis of nucleotides. Given the explosion of structural information on ABC transporters (for example, of the ~50 human ABC transporters, structures have been reported for the human A1, A4, B1, B2, B3, B4, B6, B8, B10, B11, C7, D4, G2, G5, G8 transporters in addition to D1), it is a challenge to interpret new results in the context of existing knowledge about ABC transporters. The transport assays provide an approach to connecting structure and function, but too little information is provided about the assays to make those connections. Of the ~900 mutants, why were the variants in figure 4B selected? Where are they located in the structure? What do the results of the transport assay mean? For example, is the reduction of C22:S-CoA in the cytosol for the G343V mutant due to impaired transport or to reduced expression, misfolding or some other cause? mutations are mainly mapped to the TMDs, which may affect the conformational changes during substrate transport.Please show the correct trafficking, expression (normalised to WT) and folding of all mutations tested in HEK293 cells.Please show that the purified protein is capable of turning over ATP and that the basal ATPase activity stimulated by the substrate. We would recommend carrying out complementary ATPase assays on key mutations also. It is critical to better integrate the mutagenesis data with the structure.

Point accepted. We have discussed the transport mechanism of ABC transporters in the revision based on integrative analysis of expression level, ATPase assay and trafficking. We also cited previously literatures that reported ALD-related mutants mentioned in our study. We mapped the mutations selected for transport assay on the structure as suggested. In the original manuscript, we temporarily classified the numerous ALD-associated mutations into four groups based on structural mapping of these mutations (Figure-3). Due to the complexity for preparing the lipid extraction to fulfill the transport assay, we further selected several representative mutations following the structural guidance. Additionally, we removed the peroxisomal helix and the C-terminal coil-coiled domain in ABCD1 to investigate the role of these two featured structural elements.

To interpretate the results of the transport assay, we analyzed the trafficking, expression, and the folding of these mutants through confocal microscope, quantitative Western blot, and thermo-stability assessment as the reviewers suggested. The ATPase activity of these mutations has been presented as well. (See Author response image 2)

3. Please provide some rationale basis for the additional structural features found in ABCD1 (peroxisomal helix and the C-terminal coil-coiled domain) and how this might be related to its function?

As described above, we have individually truncated the peroxisomal helix and the C-terminal coil-coiled domain and examined the trafficking, expression, and the thermo-stability of these two constructs. The ATPase activity and the transport activity of these mutations has been presented as well. Judged from the ATPase and transport activity, the peroxisomal helix exhibit obstructed ATP hydrolysis and lipid transport (decreased ~50%), whereas deletion of the C-terminal coil-coiled domain have more influences on the lipid transport (decreased ~80%) than ATP hydrolysis. One possible reason is that deletion of the C-terminal domain has influence on the stability of ABCD1 with the NBD domain unaffected. Another possibility is that the C-terminal domain facilitates the conformational change of ABCD1 during the lipid transport. This conjecture is supported by structural studies reveals that the coil-coiled domain cannot be identified from the ATP bound state.

**Author response table 1. sa2table1:** 

State	Confirmation	Peroxisomal helix	c-terminal coiled domain	Reference
Apo	Cytosol-facing	Presence	Presence	This study
Apo (chimera)	Cytosol-facing	Absence	Presence	Chen et al
C22-CoA bound (chimera)	Intermediate	Presence	Absence	Chen et al
ATP bound (Chimera)	Peroxisomal-facing	Absence	Absence	Chen et al
Oleoyl-CoA bound	Cytosol-facing	Presence	Presence	Wang et al
ATP bound	Peroxisomal-facing	Absence	Absence	Wang et al
C26 bound	Cytosol-facing	Presence	Presence	Xiong et al
C26-CoA & ATP bound	Cytosol-facing	Presence	Presence	Xiong et al
ATP bound	Peroxisomal-facing	Absence	Absence	Xiong et al
ATP-γS bound state 1	Peroxisomal-facing	Absence	Absence	Le et al
ATP-γS bound state 2	Cytosol-facing	Presence	Absence	Le et al

The peroxisomal helix becomes disordered in the presence of ATP. This conformational change led to the lipid-binding cavity open to the peroxisome lumen. The peroxisomal helix connects TM5 and TM6. TM5 is coordinated by the NBD of another protomer whereas TM6 is proceeded by NBD. Deletion of the peroxisomal helix may lead to spatial constraints for the movement of TM5 and TM6 towards the symmetry axis, finally transduced to the NBD.

**Author response image 3. sa2fig3:** 

4. The final model is difficult to follow and this, of course, relates to the "actual" location of the substate-binding site. Is an outward-facing conformation to be expected in ABCD1? Furthermore, the conformational changes as proposed by comparing to AlphaFold models should be made with caution until experimental validation of these states. Please incorporate the AlphaFold models more carefully into the paper, i.e.,what are the state-specific co-evolved residue-pairs predicted in the AlphaFold models, do these make sense, and can these contact pairs be experimentally validated by mutagenesis?

As we mentioned above (major point 1), we have rewritten the description of the potential binding site in our model to avoid controversial interpretation on the binding site of VLCFA-CoA. We found that the lipid density in our model partially overlaps with the acyl tail of C22-CoA, identified by Chen and co-workers. Further, we compared our structure and the reported ALDP models. Our structure is similar to the apo-state by Chen et al. and the Oleoyl-CoA bound state by Wang et al. Similarly, both the apo and oleoyl-CoA bound state exhibit cytosolic-facing conformation. Based on the structure alignment and functional state of our structure, we consider our state is cytosolic-facing conformation.

For the AlphaFold models in the original paper, we used the model from the AlphaFold Protein Structure Database for structure comparison. The database provides monomer structure model rather than multimer. As suggested by the reviewers, we provide the dimeric model through AlphaFold predicted on local-computer (see Author response image 4). This process follows the general command of AlphaFold by providing the sequence file. However, we are inaccessible to read out the state-specific co-evolved residue-pairs used by AlphaFold at the current stage. Instead, we performed the co-evolved residues by another web-server developed by David Baker group (http://gremlin.bakerlab.org/). We provided the results along with the revised manuscript. More co-evolved residues pairs are located at the NBDs than TMDs (see Author response image 5).

**Author response image 4. sa2fig4:** 

**Author response image 5. sa2fig5:** 

The predicted ALDP-dimer is reminiscent of the ATP-bound state. Such result is not surprising because the majority conformation of multiple ABC transporters has been determined to be the ATP-bound states [ref]. Before the cryo-EM structures of ALDP from the several independent groups, the closest homolog of ABCD1 with structure is ABCD4, which displays an ATP-bound conformation. Structural alignment between our model and the predicted or the ATP-bound state reveals dramatic conformational changes at the TMD and NBD.

5. The paper needs to be extensively re-written. We would like to see how the structure uncovers new mechanistic insights. With the additional functional data requested, consider how the ABCD1 structure enables a understanding of why very long chain fatty acids use such an ABC transporter structures for being imported into peroxisomes. How does this differ to other ABC transporters that transport lipids and why?

We compared the differences of the binding cavities between ALDP and ABCB4. ABCB4 is a phosphotidylcholine transporter. Typically, ALDP consists of two binding pockets listed in the large cavity formed by TMDs. ABCB4 harbors one lipid binding pocket. The cavity of ALDP is larger than ABCB4. The different binding manner of these two ABC transporters is associated with distinct conformational changes during lipid transport. The two protomers of ALDP moves away from the cavity center in the presence of ATP. These differences may be attributed to two reasons: (1) ALDP is assembled by two identical protomers whereas ABCB4 is assembled by single chain polypeptides; (2) The chemical shape of the specific substrates. Unlike most fatty acids, VLCFAs are too long to be metabolized in the mitochondria and must be metabolized in peroxisomes. We use C22:0 lipids for the comparison. C22:0-CoA has one acyl tail and a much larger hydrophilic head in contrast to C22:0-PC.

**Author response image 6. sa2fig6:** 

Reviewer #1 (Recommendations for the authors):1. The modelled two putative Co-A esters lack the positively charged residues for coordinating phosphates. Indeed, the Co-A binding in the human ABCD1 structure by Chen and co-workers make more chemical sense https://www.biorxiv.org/content/10.1101/2021.09.24.461565v2.full.pdf. This study found they same shaped density as you seen in your ABCD1 structure, but have modelled the lipid 18:0 Lyso PE molecule into this location instead. Judging the deposited maps I think that this is the more likely lipid for your density, rather than CoA as proposed. Please read this submission more carefully and update accordingly as I find it worrying that this has not been picked up.

Point accepted. Please see “Essential Revisions (for the authors)” Major point 1.

2. Please check that ABCD1 mutations are trafficked correctly to peroxisomes. Also one needs to normalise uptake against expression level. Please complement substrate uptake experiments with ATPase activity measurements also.

Point accepted. Please see “Essential Revisions (for the authors)” Major point 2.

3. This paper is difficult to follow in places and using AlphaFold structures to make conclusions regarding local conformational changes seems premature at this stage without further validation.

Please see “Essential Revisions (for the authors)” Major point 4.

4. Please provide some rationale basis for the additional structural features found in ABCD1 (peroxisomal helix and the C-terminal coil-coiled domain) and how this might be related to its function?

Please see “Essential Revisions (for the authors)” Major point 3.

We thank this reviewer for his/her thoughtful suggestions which we fully accepted.

Reviewer #3 (Recommendations for the authors):The CoA-ester binding should be discussed in more details.Although several mutants have been prepared, they are not really discussed in the context of the manuscript or previously published work. The discussion needs to expand further. Certain assumptions have been made but there is no references or experimental validation from this work to warrant such conclusions; eg p6 'Pathogenic mutations on these residues may disturb the stability of ALDP. The residues of the third group are located on the NBDs, which may hinder the hydrolysis of nucleotides. The other mutations are mainly mapped to the TMDs, which may affect the conformational changes during substrate transport.'. Based on their structure it should be possible to explain better why and how the mutants may affect the function.The authors should show that the purified protein is capable of turning over ATP and that the basal ATPase activity stimulated by the substrate.

Point accepted. We have discussed the mutants in the revision based on the supplement of ATPase activity assay, trafficking information, thermo-stability assessment.

Figure 4B; are all mutants expressing at similar levels as the WT protein? They authors should show a Western Blot. Additionally, what do the error bars represent?

Point accepted. We have shown the WB results of the expression level in the revision.

The authors should briefly state in the M and M how they prepared the Alphafold models; ie collar or full Alphafold. Why not use the Alphafold multimer and make the homodimer?

Please see “Essential Revisions (for the authors)” Major point 4.

[Editors’ note: further revisions were suggested prior to acceptance, as described below.]

– The main impact of the paper is the ALDP structure with a number of supporting mutations. The current Manuscript word length is around 1800 words and 4 figures, which is close to the requirements for a short report (1500 words and 3 figures). We think the paper would come across much better as a short report, and ask that you please re-write to meet these requirements and improve the readability of the text in the process.

Thanks for the kind suggestions. To comply with your editorial requests, the revised manuscript is within 1500 words and 3 figures. We thoroughly rewritten the main text following the review comments and suggestions.

The referees have further concerns to be addressed:– line 100: "The TM2 serves as a mechanical lever: the cytosolic side moves towards the central cavity, while the peroxisomal side may move backwards the symmetry axis"Two points:i. is TM2 really a mechanical level? A lever is a rigid object with a fulcrum point that amplifies an input force to become an output force. Is this what is meant? If so, where is the fulcrum and what are the input and output forces?ii. I don't understand what is being described in the second part of this sentence.

Point accepted. We have removed the “mechanical level” as this description is not accurate.

– Line 148 "Intriguingly, there are four constructs exhibit unaffected or little enhanced ATPase hydrolysis but obstructed lipid transport"

To discriminate the disease-derived mutants and the rational designed mutants, we have rewritten this paragraph and removed the subjective vocabulary like “Intriguingly”. Please see page 7 in the revised manuscript.

– Line 164 " Truncation of the CH does not change the completeness of the NBD"What is meant by the completeness of the NBD?

We keenly accept the suggestion from this reviewer and rewrite this sentence as below:

“Truncation of CH (Δ683-745) does not change the assembly of NBD but affects the conformational changes between the two protomers”

– Line 194 – what precisely do these dimensions (42 Å or 40 Å) describe? – this implies that the TMDs have similar dimensions to accommodate their substrates – but lines 203 (and the abstract) state that the binding pocket of ALDP is much larger than ABCB4. Please clarify.What are the volumes of the binding cavities in ALDP and ABCB4, and how does the volume of PC compare to the volume of the VLCFA substrate?

Point accepted. The dimensions of *42 Å or 40 Å* imply the TMDs have similar dimensions. But ALDP has two substrate binding pockets and two exits for substrate release, while ABCB4 coordinates one PC molecule. In the revised manuscript, we have rewritten this part. Because the both VLCFA and PC are flexible, we temporarily compared the molecule size through describing the molecular weight and roughly measured the length of these two molecules in revised Figure 3 —figure supplement 5. The volumes of one binding cavity is ~1695 Å^3^ for C22:0-CoA and ~1236 Å^3^ for PC.

– Line 264 – Figure 4 – which PDBs are which? (ie ligand-free, substrate-bound, ATP-bound)?

Point accepted. We have labeled the states of corresponding PDB files.

– One last point that likely reflects an issue on my end, but I was not able to see the EM map postrun9.mrc in coot 0.9.6 – Coot opened the map and provided a histogram of the density, but it didn't display anything over the "A_fit_real_space_refined-coot-34_real_space_refined_001_all_states.pdb" coordinates.

We have provided the map and coordinate again in this revision. Please see attached files.

– Why is the structure in line 190 compared to ABCB4 and not the other structures listed in line 90? It makes more sense to discuss the mechanism based on the available structures of the same family in addition to the ABCB4.

Point accepted. We have compared the structures of ALDP in different conformations as suggested. In the last version, we wanted to compare the ABC transporters of different long-tail lipids, thus we selected the structures of ALDP and ABCB4. This part has been moved to the supplementary figures.

– It is still unclear how they selected their mutants; lines 122-125 do not have any refs of previously identified mutations. Do we know any of them being specific disease related?

Point accepted. We have rewritten this part to describe how these mutants have been selected. The references have been cited properly as suggested. Briefly, we selected six ALD-derived mutants that localized at the substrate binding site, the NBD and the C-terminal coiled-coil domain. These mutants are W339R, S342P, G343V, A396T, Q544R, and T693M. Additionally, we designed four constructs based on the structural findings, including S164A/Y310A, E291A/R518A, Δ364-374 to delete the peroxisomal helix and Δ683-745 to truncate the whole C-terminal coiled-coil domain. These four mutants are designed for probing how the associated structure elements affect the activity of ALDP.

– Lines 85-91 contradict each other; they state that there is lipid-like density but at the same time they write that they have an apo structure.

Point accepted. We compared the structures of ALDP from different groups. Chen et al. also reported two lipid-like densities in the apo-state which have been built as PE in their model. Nonetheless, to avoid inaccurate interpretations, we have change the description as unknown densities.

Reviewer #2 (Recommendations for the authors):The structure of human ALDP is of significance given the role of this transporter in the transport of VLCFAs across the peroxisomal membrane. Nearly a thousand ALD-causing mutations have been identified for ALDP and the structure provides a framework for assessing their functional consequences. In this revised manuscript, the authors have addressed many of the reviewers' concerns in the rebuttal letter. Given the importance of this transporter, particularly the human version of ALDP, I am sympathetic to the publication.The manuscript, however, needs extensive polishing to be suitable for publication, however. Here are a few examplesline 100: "The TM2 serves as a mechanical lever: the cytosolic side moves towards the central cavity, while the peroxisomal side may move backwards the symmetry axis"Two points:i. is TM2 really a mechanical level? A lever is a rigid object with a fulcrum point that amplifies an input force to become an output force. Is this what is meant? If so, where is the fulcrum and what are the input and output forces?ii. I don't understand what is being described in the second part of this sentence.

Please see above.

Line 148 "Intriguingly, there are four constructs exhibit unaffected or little enhanced ATPase hydrolysis but obstructed lipid transport"

Please see above.

Line 164 " Truncation of the CH does not change the completeness of the NBD"What is meant by the completeness of the NBD?

Please see above.

line 194 – what precisely do these dimensions (42 Å or 40 Å) describe? – this implies that the TMDs have similar dimensions to accommodate their substrates – but lines 203 (and the abstract) state that the binding pocket of ALDP is much larger than ABCB4. Please clarify.What are the volumes of the binding cavities in ALDP and ABCB4, and how does the volume of PC compare to the volume of the VLCFA substrate?

Please see above.

line 264 – Figure 4 – which PDBs are which? (ie ligand-free, substrate-bound, ATP-bound)?

Please see above.

One last point that likely reflects an issue on my end, but I was not able to see the EM map postrun9.mrc in coot 0.9.6 – Coot opened the map and provided a histogram of the density, but it didn't display anything over the "A_fit_real_space_refined-coot-34_real_space_refined_001_all_states.pdb" coordinates.

Please see above.

We thanks this reviewer for his/her constructive suggestions.

Reviewer #3 (Recommendations for the authors):The authors have significantly improved the manuscript with the addition of additional functional data of several mutants.A few more points to strengthen the manuscript:Why is the structure in line 190 compared to ABCB4 and not the other structures listed in line 90? It makes more sense to discuss the mechanism based on the available structures of the same family in addition to the ABCB4.

Please see above.

It is still unclear how they selected their mutants; lines 122-125 do not have any refs of previously identified mutations. Do we know any of them being specific disease related?

Please see above.

Lines 85-91 contradict each other; they state that there is lipid-like density but at the same time they write that they have an apo structure.

Please see above.

The grammar in several sections needs to be improved.

We have rewritten the manuscript and examined the grammars. We keenly thank this reviewer for his/her constructive suggestions.